# Arterial Blood Gas Analysis and Clinical Decision-Making in Emergency and Intensive Care Unit Nurses: A Performance Evaluation

**DOI:** 10.3390/healthcare13030261

**Published:** 2025-01-28

**Authors:** Arian Zaboli, Chiara Biasi, Gabriele Magnarelli, Barbara Miori, Magdalena Massar, Norbert Pfeifer, Francesco Brigo, Gianni Turcato

**Affiliations:** 1Innovation, Research and Teaching Service (SABES-ASDAA), Teaching Hospital of the Paracelsus Medical Private University (PMU), 39100 Bolzano, Italy; magdalena.massar@sabes.it (M.M.); francesco.brigo@sabes.it (F.B.); 2Intensive Care Unit, Hospital of Bolzano (SABES-ASDAA), 39100 Bolzano, Italy; chiara.biasi@sabes.it (C.B.); barbara.miori@sabes.it (B.M.); 3Department of Emergency Medicine, Hospital of Merano-Meran (SABES-ASDAA), 39012 Merano, Italy; gabriele.magnarelli@sabes.it (G.M.); norbert.pfeifer@sabes.it (N.P.); 4Department of Internal Medicine, Intermediate Care Unit, Hospital Alto Vicentino (AULSS-7), 36014 Santorso, Italy; gianni.turcato@yahoo.it; 5Department of Health Sciences, UniCamillus—Saint Camillus International University of Health and Medical Sciences, 00131 Rome, Italy

**Keywords:** blood gas analysis, critical care nursing, intensive care units, emergency nursing, arterial blood gas, decision-making, critical illness, clinical competence

## Abstract

**Background:** This study aimed to evaluate Emergency Department and Intensive Care Unit nurses’ skills in interpreting blood gas analysis results and to use those interpretations in clinical decision-making. **Methods:** In this prospective, multicenter, simulation-based study, nurses from the Emergency Department (ED) of Merano Hospital and the Intensive Care Unit (ICU) of Bolzano Hospital, Italy, were presented with 16 clinical vignettes based on real patient cases. These vignettes were designed to evaluate the nurses’ ability to identify patients with time-dependent conditions and recommend appropriate therapeutic interventions. Outcomes measured included sensitivity, specificity, and agreement with physician-assigned urgency levels and therapy recommendations. **Results:** Among the 43 participants (26 ICU and 17 ED nurses), specificity in excluding patients without time-dependent conditions or organ replacement needs was high. However, sensitivity in identifying time-dependent conditions was less than 50%. Agreement with physician-assigned urgency levels was low, with Cohen’s kappa values of 0.139 for ICU nurses and 0.218 for ED nurses. Nurses with lower self-confidence in interpreting BGA results made more errors, while other personal or professional factors did not significantly impact performance. **Conclusions:** Although critical care nurses can effectively rule out patients without time-dependent conditions, their ability to identify such conditions requires improvement. These findings underscore the need for targeted training programs to enhance nurses’ BGA interpretation skills and clinical decision-making in high-pressure, time-sensitive situations.

## 1. Introduction

In recent years, blood gas analysis (BGA) has gained increasing prominence in clinical practice, becoming an indispensable tool in the management of critically ill patients [1]. It plays a pivotal role in the care of a diverse range of urgent and time-dependent conditions, as underscored by numerous clinical guidelines [2,3,4]. Time-dependent conditions are those conditions with a high risk of progression that pose a significant threat to patient outcomes and demand rapid identification and intervention. The effective management of these time-dependent conditions hinges on clinical decision-making—a structured process in which healthcare providers, including physicians and nurses, interpret clinical data, such as BGA results, to inform prompt and accurate treatment decisions. This process requires the integration of clinical expertise, situational awareness, and evidence-based judgment to prioritize interventions that mitigate risks and optimize patient outcomes.

For example, BGA is routinely recommended for patients presenting with major bleeding, respiratory distress, sepsis, and other critical conditions [2,3,4,5]. Consequently, it has become a cornerstone of care in emergency departments (EDs) and intensive care units (ICUs).

Despite its widespread use and established clinical utility, interpreting BGA results remains a complex task [6]. No standardized, universally straightforward approach exists, as interpretation depends on multiple factors, including the patient’s acute condition, chronic health status, ongoing treatments, comorbidities, and baseline characteristics [6,7,8]. While BGA provides rapid and valuable diagnostic insights, the intricacy of its interpretation continues to challenge healthcare providers [7,8].

Nurses play a critical role in the application of BGA, performing the analysis both under medical supervision and autonomously to monitor and evaluate a patient’s clinical condition [9]. To fulfill this role more effectively and with greater insight, nurses should be able to interpret BGA results accurately and identify critical changes in a patient’s status [9]. Although the literature offers various methodologies to support critical care nurses in interpreting BGA, clinical studies specifically evaluating their interpretative skills remain scarce [10,11,12]. This clinical challenge reflects a dual reality: while nurses are often required to interpret and act upon BGA results in their daily practice, their actual competencies in this area remain largely unexplored. This knowledge gap highlights the need to determine whether nurses are effectively using BGA data to inform clinical decision-making and modify patient management. To address this, the present study aimed to evaluate the BGA interpretive skills of nurses working in EDs and Intensive Care Units (ICUs) through simulated clinical scenarios. By assessing their ability to analyze BGA results and identify time-dependent conditions, the study seeks to identify the educational and training needs of critical care nurses, with the goal of enhancing their clinical decision-making capabilities.

The study was therefore designed with the following aims:To assess the ability of ICU and ED nurses to interpret BGA results accurately, with a focus on identifying time-dependent conditions.To compare the performance of both nursing groups in interpreting clinical urgency levels and therapeutic interventions.To identify areas of training improvement by analyzing the influence of personal and professional factors, including self-confidence in BGA interpretation.

The study objectives were identical for both ICU and ED nurses, although differences in their baseline competencies were anticipated due to the nature of their work environments.

## 2. Methods

### 2.1. Design

This study was conducted as a prospective, multicenter, simulation-based investigation between 1 May 2024 and 31 October 2024.

### 2.2. Study Setting and Sampling

The participating sites included the ED at Merano Hospital and the ICU at Bolzano Hospital, both located in Italy.

The ED at Merano Hospital is the second largest in the province and is equipped with a point-of-care BGA device, ensuring rapid access to BGA results for critically ill patients. Similarly, the ICU at Bolzano Hospital serves as the provincial reference center for managing critically ill adult patients at the regional level and is equipped with a dedicated BGA device to support the complex care needs of its patients.

This setup ensured that both study sites were well-resourced, with the necessary technology to integrate BGA into routine critical care workflows.

To develop the clinical vignettes used in this study, 16 real-life cases were anonymously extracted from the electronic database of the ED at Merano Hospital. The case selection process was carried out by two senior physicians, each with over 10 years of experience: one specializing in emergency medicine within the ED and the other in intensive care within the ICU. The selection process was designed to include a balanced mix of acute, time-sensitive cases and less severe scenarios to reflect the diverse clinical presentations encountered in critical care settings. Although no formal validation was carried out, this process aimed to capture a diverse range of acute and time-sensitive conditions, with an emphasis on scenarios in which BGA results were pivotal to decision-making.

The vignettes were structured to include the following information: (a) patient age; (b) patient sex; (c) a concise medical history; (d) vital signs; and (e) complete arterial BGA results, presented in their entirety as they would appear in clinical practice, without highlighting abnormal values or providing reference ranges. A sample clinical vignette is available in Appendix A. The vignettes were created by integrating the clinical case, vital signs, and BGA to closely reflect real-world clinical settings. In everyday clinical practice, nurses do not rely solely on BGA data but also assess the patient’s overall condition, including vital signs, to guide their clinical decisions. This approach aimed to provide a more accurate and comprehensive simulation of the clinical environment.

For each vignette, the two physicians assigned a clinical urgency level based on the BGA findings and other clinical data, using a standardized three-tier classification system:

Non-Urgent: Patients not requiring evaluation by an ED or ICU physician based on their clinical presentation and arterial BGA results.

Urgent: Patients requiring evaluation by an ED or ICU physician within 60 min of obtaining the arterial BGA.

Emergency: Patients requiring immediate evaluation due to the severity of their clinical presentation and arterial BGA results.

These levels of clinical urgency were not based on standardized protocols or a triage system, but were instead derived from the clinical assessment made by two physicians who consulted with each other to determine the patient’s severity.

The physicians reviewed all available patient records, including data from the ED and subsequent hospitalizations, to determine the presence of time-dependent medical conditions. This evaluation utilized all relevant information contained in the database to ensure comprehensive case analysis.

Lastly, the physicians assessed the need for therapeutic interventions following BGA, categorizing them as
(1)Non-Invasive Therapies: Treatments such as fluid resuscitation or oxygen therapy (assessed using a dichotomous response: yes/no).(2)Organ Replacement Therapies: Interventions such as invasive mechanical ventilation or dialysis (also assessed using a dichotomous response: yes/no).

### 2.3. Inclusion and Exclusion Criteria

In the participating departments, nurses working autonomously in both ICU and ED settings were identified as eligible for the study. Only those capable of independently managing all aspects of patient care were included, excluding nurses currently in mentorship or training phases. In the ED, eligibility was restricted to nurses assigned to high-complexity areas responsible for managing critically ill patients with advanced care needs. For this reason, a convenience sampling method was used for the study, a common approach in this type of research.

A formal sample size calculation was not performed for this study, as it was designed as a descriptive, simulation-based evaluation of nurses’ BGA interpretation skills. The sample was determined based on the availability and willingness of eligible ICU and ED nurses to participate. While a larger sample could enhance statistical power, the 43 nurses included were deemed sufficient to achieve the study’s exploratory objectives. The high participation rates (92.8% in ICU, 73.9% in ED) further support the representativeness of the sample for the specific departments involved.

The enrolled nurses were not required to undergo any specific training or structured framework; rather, their clinical competence, as routinely applied in their daily practice, was evaluated.

### 2.4. Data Collection

Participation in the study was voluntary, with no penalties for non-participation. Recruitment was conducted through emails sent to staff members and presentations during departmental meetings. To further encourage participation, the data collection process was fully anonymized, with no personal identifiers (e.g., names, surnames, or dates of birth) recorded. Participation in the study was entirely voluntary, and none of the participants were obligated to take part. Nurses were invited to express their interest in participating by contacting the local investigators at the two departments, either via email or verbally, in accordance with current Italian regulations and the requirements of the local ethics committee. To ensure participant confidentiality, data collection was conducted anonymously.

Upon providing verbal consent, participants completed the study questionnaire in a supervised environment monitored by a local investigator. All nurses were required to complete the questionnaire in a designated room under the supervision of a local investigator. The use of electronic devices or any form of external communication was strictly prohibited during the questionnaire session. This measure was implemented to minimize external influences and ensure the integrity of the study process. This setup ensured that no external aids, such as electronic devices, reference materials, or consultations, were used when completing the clinical vignettes. Participation time was recognized by the local healthcare organization as part of the nurses’ official working hours.

Before completing the 16 clinical cases, each nurse provided personal and professional background information. The following data were collected:(a)Sex and age (in years);(b)Total years of nursing experience;(c)Years worked in the current department;(d)Personal interest in BGA, rated on a scale from 0 (no interest) to 10 (maximum interest);(e)Self-reported engagement in personal study of BGA (yes/no);(f)Self-assessed confidence in performing arterial puncture, rated on a scale from 0 (no confidence) to 10 (maximum confidence);(g)Self-assessed confidence in interpreting BGA results, rated on a scale from 0 (no confidence) to 10 (maximum confidence);(h)Frequency of performing BGA during a shift, rated on a scale from 0 (never) to 10 (more than 10 times per shift);(i)Participation in a BGA-related training course within the past five years (yes/no);(j)Highest degree obtained, categorized as Bachelor’s in Nursing; Clinical Master’s Degree (a specialized post-graduate nursing qualification in Italy); or Master of Science in Nursing.

This approach ensured a comprehensive understanding of the participants’ professional background and its potential influence on their performance in the study.

### 2.5. Study Endpoints

The primary outcome of the study was to evaluate nurses’ ability to accurately identify patients with time-dependent conditions, as determined by two experienced physicians who had full access to the patients’ complete clinical documentation.

The secondary outcome focused on assessing the nurses’ ability to identify patients requiring non-invasive therapies and organ replacement therapies. Their decisions were compared to those made by the physicians, serving as the reference standard.

This study did not prespecify a specific performance level, as it was descriptive in nature. Instead, the focus was on measuring the nurses’ ability to identify time-dependent conditions and exclude patients without urgent medical needs based on their interpretation of BGA results. Hence, we assessed overall performance across both nursing groups without setting predefined benchmarks.

### 2.6. Data Analysis

The characteristics of the enrolled nurses were compared between the two groups (ICU and ED) using univariate analyses. Continuous variables were reported as mean ± standard deviation (SD) or median with interquartile range (IQR), depending on the distribution of the data. Univariate comparisons were conducted using Student’s *t*-test, the Mann–Whitney U test, or the Kruskal–Wallis test, as appropriate. Categorical variables were expressed as absolute frequencies and percentages and compared using Fisher’s exact test or the Chi-squared test.

To compare the urgency classifications assigned by ICU and ED nurses with those of the physicians, a 3 × 3 contingency table was constructed. Cohen’s unweighted kappa coefficient, along with its 95% confidence interval (95% CI), was used to assess the level of agreement between nurses and physicians.

For the secondary outcomes, which involved the identification of patients requiring non-invasive therapies or organ replacement therapies, 2 × 2 contingency tables were constructed. Sensitivity, specificity, positive predictive value (PPV), negative predictive value (NPV), and their respective 95% CIs were calculated for comparisons of nurses’ evaluations with those of the physicians.

For the primary outcome—the identification of patients with time-dependent medical conditions—2 × 2 contingency tables were also used to compare nurses’ evaluations with those of the physicians. Similar statistical measures (sensitivity, specificity, PPV, NPV, and 95% CI) were calculated, and Cohen’s unweighted kappa coefficient with its 95% CI was used to evaluate agreement.

Additionally, the average number of errors made by nurses in identifying time-dependent medical conditions was analyzed. Nurses were divided into two groups: those who made more errors than the average and those who made fewer. Personal and professional characteristics were compared between these two groups to explore potential correlations with error frequency. The same univariate comparison methods described above were applied.

A *p*-value of <0.05 was considered statistically significant for all analyses. Statistical analyses were performed using STATA software, version 16.1.

### 2.7. Ethical Considerations

Ethical review and approval were waived for this study due to this being a simulation study. The study was conducted in compliance with the local ethics committee guidelines (Comitato Etico per la Sperimentazione Clinica, Azienda Sanitaria dell’Alto Adige, Bolzano, Italy, protocol number Prot.0153571-BZ) and adhered to the Declaration of Helsinki on Ethical Principles for Medical Research Involving Human Subjects. As this was a simulation study, written consent from the nurses was not required; participation was voluntary and based on their expressed interest.

## 3. Results

Among the eligible ICU nurses, 92.8% (26/28) chose to participate in the study, while 73.9% (17/23) of eligible ED nurses agreed to participate. This resulted in a total of 43 nurses being enrolled in the study, comprising 60.5% (26/43) ICU nurses and 39.5% (17/43) ED nurses.

The demographic and professional characteristics of the participating nurses are summarized in Table 1.

ICU nurses were significantly older than ED nurses (*p* = 0.021) and had more years of nursing experience (*p* = 0.022). In contrast, ED nurses reported higher self-confidence in interpreting BGA results, with a mean confidence score of 6.3 compared to 5.3 for ICU nurses (*p* = 0.028). However, ED nurses performed BGA less frequently during their shifts, with a median frequency of 8 compared to 10 for ICU nurses (*p* < 0.001).

Table 2 provides the contingency table comparing the urgency levels assigned by the nurses to those assigned by the physicians.

As shown in Table 2, ICU nurses more frequently categorized patients as “emergency” compared to ED nurses. The analysis of Cohen’s kappa coefficients reveals low agreement between nurses and physicians in both groups. For all nurses combined, the overall Cohen’s kappa coefficient for urgency level assignment was 0.169 (95% CI: 0.127–0.213), indicating poor agreement with the physicians’ classifications and a significant gap in the ability to assess clinical urgency accurately. The low consistency between nurse assessments and physician urgency levels may indicate that, despite the familiarity with clinical data, nurses still face challenges when integrating complex information such as BGA results, patient history, and vital signs into cohesive clinical decisions.

Table 3 presents the 2 × 2 contingency tables for the secondary outcomes, comparing the nurses’ assessments of the need for non-invasive and organ replacement therapies to those determined by the physicians.

As shown in Table 3, both ICU and ED nurses demonstrated high sensitivity when assessing the need for non-invasive therapy, albeit at the expense of lower specificity. In contrast, when evaluating the need for organ replacement therapy, both groups of nurses exhibited high specificity, accompanied by lower sensitivity. Across all scenarios, a high NPV was observed.

Regarding the primary outcome of the study—the identification of patients with time-dependent medical conditions—both ICU and ED nurses achieved high specificity, as reported in Table 4. However, sensitivity remained low in both groups, consistently falling below 50%. The low sensitivity observed across both groups suggests that while nurses can exclude patients without time-dependent conditions, they may not be sufficiently trained or confident in identifying more subtle or less obvious cases.

An analysis of discrepancies in identifying patients with time-dependent medical conditions revealed that, on average, each nurse exhibited four discrepancies in their evaluations compared to the physicians’ assessments. Using this value as a cut-off, nurses were divided into two groups: those who made more than four errors and those who made four or fewer errors.

The characteristics of nurses in these two groups were compared to explore factors potentially associated with higher error rates. The results of this comparison are detailed in Table 5.

As presented in Table 5, no specific nurse characteristics, such as age, years of experience, or educational background, were found to be significantly associated with a higher number of errors. The sole statistically significant factor was confidence in interpreting BGA results. Nurses who made more errors reported significantly lower self-confidence in their interpretative abilities compared to those who made fewer errors (*p* = 0.011). This finding highlights the importance of self-assessed competence in influencing performance.

## 4. Discussion

This study demonstrated that critical care nurses are proficient at confidently excluding patients without time-dependent medical conditions, showing high specificity in identifying both time-dependent medical conditions and the need for organ replacement therapies. However, their overall performance remains suboptimal, particularly in terms of agreement with physicians regarding urgency levels in clinical cases.

In this study, it was necessary to address the concept of patient urgency in order to define the degree of clinical severity. This step was essential because nurses are not responsible for making diagnoses, and a surrogate measure of severity, such as clinical urgency, had hence to be identified. Nurses working in acute care settings should be able to differentiate between the most critical patients and those with less severe conditions, allowing them to promptly initiate the necessary interventions to safeguard patient health. While clinical urgency may not be a perfect choice, it was considered the most practical and applicable solution for this study, given the diverse settings and roles of the nurses involved.

The findings carry significant practical implications. The existing literature, which is often outdated, largely focuses on theoretical algorithms or methods for interpreting BGA results without evaluating nurses’ actual skills or pinpointing their knowledge gaps [9,11,12,13]. Moreover, it is well recognized that BGA results are influenced not only by patient-related factors but also by ongoing therapies and the patient’s acute and chronic conditions [14,15,16,17]. This complexity makes the idea of a universally applicable flowchart or standardized method impractical across all clinical settings [15,16,17].

Unlike diagnostic tools such as electrocardiograms (ECGs), which are standardized and comparatively straightforward to interpret, BGA interpretation involves a higher degree of complexity [18]. Factors that may be evident to ICU nurses might not be as clear to ED nurses, and vice versa. Consequently, this study did not aim to assess interpretive competency at a granular level but instead focused on the broader clinical impact of nurses’ decision-making. It sought to determine whether nurses could evaluate the severity and urgency of a patient’s condition based solely on a brief medical history, vital signs, and arterial BGA results. The findings indicate that nurses are not yet equipped to independently interpret BGA at even a macro level, underscoring the need for further investigation into both general and detailed competencies. A deeper understanding of these skills could lead to the development of practical tools that are genuinely useful in clinical practice.

For critical outcomes, such as identifying time-dependent medical conditions and determining the need for organ replacement therapy, the study reported high specificity. This finding is reassuring, as it suggests that nurses are effective at ruling out patients who do not require urgent interventions. Clinically, this is often more valuable than achieving high sensitivity, as erring on the side of alerting physicians—even when unnecessary—may be preferable to the risk of missing a critical case [19]. This approach aligns with the principles of clinical prediction tools like the National Early Warning Score (NEWS), which prioritize high specificity [20,21,22]. Such tools excel at identifying low-risk patients who are unlikely to deteriorate, making them reliable for clinical application. Similarly, this study indicates that nurses are adept at identifying patients who do not present with time-dependent conditions, even in high-stakes scenarios. Nonetheless, their overall performance remains insufficient, emphasizing the need for robust training programs to enhance their competencies.

An interesting finding in this study was the significant differences in work experience between ICU and ED nurses. ICU nurses, on average, were older, had more years of total nursing experience, and had worked longer in their current department compared to their ED counterparts. These disparities could have potentially influenced the study results, as clinical experience often correlates with higher competence in managing complex situations. However, despite these differences, the performance outcomes between ICU and ED nurses in identifying time-dependent conditions and interpreting BGA results were remarkably similar.

This finding highlights the importance of ongoing education and specialized training, even for experienced personnel. While experience is undoubtedly valuable, it may not fully compensate for the nuanced knowledge and interpretative skills required in critical care environments [23,24]. Interestingly, similar trends have been frequently observed in other domains and areas, such as triage, ECG interpretation, and cardiopulmonary resuscitation (CPR), where more experienced personnel often perform at a level comparable to their less experienced counterparts [23,24,25]. This reinforces the need for continuous training in all critical care areas.

These observations underscore the value of implementing a periodic training approach, similar to the current standard for CPR recertification every two years or less [25,26]. Expanding such an approach to include topics like BGA interpretation could further enhance clinical competencies among nurses, ultimately improving patient safety and outcomes.

An important finding is that the only characteristic significantly associated with a higher number of errors in identifying time-dependent or non-time-dependent medical conditions was nurses’ self-perceived confidence in interpreting BGA results. Nurses with lower confidence were more likely to make errors, suggesting that self-assessment of competency is a reliable indicator of performance. This contrasts with studies in other areas, such as ECG interpretation and triage, where factors like years of experience often correlate more strongly with error rates [23,24]. These differences highlight the need for further research to identify specific factors that could help target training efforts to populations most in need of additional support in BGA interpretation.

The study has a few limitations. First, the study was conducted in a simulated environment, which, while controlled, may not fully capture the complexities and dynamics of real-world clinical practice. In actual settings, nurses have access to a broader range of patient information, including continuous monitoring data and input from multidisciplinary teams, which can influence their decision-making processes. Furthermore, the absence of real-time patient interaction in the simulation may limit the applicability of the findings to clinical scenarios where non-verbal cues and bedside observations play a crucial role in decision-making. Despite these limitations, significant efforts were made to enhance the realism of the simulation. Actual clinical cases were used to design the scenarios, and comprehensive contextual information was provided to the participants. Nevertheless, the inherent differences between simulation-based assessments and real-world practice must be considered when interpreting the results. Future studies could explore hybrid approaches that combine simulations with on-the-job evaluations to provide a more holistic understanding of nurses’ competencies in actual clinical environments. Second, the study did not evaluate detailed interpretive skills for individual BGA parameters. This was intentional, as the focus was on how BGA interpretation is applied within the broader clinical context, particularly in decision-making for patient management. Future studies could address these limitations to further refine understanding of and training in BGA competencies among critical care nurses. Third, the study was conducted in only two departments of two Italian hospitals, which limits the generalizability of the results to the broader nursing context. Some of the competencies analyzed may not be applicable in other settings.

Finally, although the sample size was relatively small, it provided valuable insights into the practice within the specific settings of the ICU and ED, as the proportion of nurses who chose to participate was high among eligible nurses. The high participation rate, particularly among ICU nurses (92.8%), reflects the real-world clinical practice and nursing capabilities within these two departments. Although the sample size limits the generalizability of the findings to broader populations, the study still offers useful data for understanding the relative strengths and areas of improvement in BGA interpretation and clinical decision-making in these critical care environments. A larger sample size would be beneficial in future studies to strengthen the external validity of these findings.

Therefore, future studies involving larger cohorts and multiple hospitals would be necessary to enhance the external validity and provide a more comprehensive understanding of nursing competencies across diverse clinical environments. Furthermore, specifying an optimal performance level could be useful for future research, particularly in terms of setting benchmarks for BGA interpretation. Achieving higher levels of accuracy in this area would enhance patient safety by ensuring that critical conditions are promptly identified and managed, improving overall departmental performance and clinical outcomes.

Targeted educational programs are essential for improving nurses’ skills in interpreting BGA results and recognizing time-dependent conditions. Specific interventions could include tailored simulation-based training sessions that replicate high-stakes scenarios commonly encountered in ED and ICU settings. These sessions should focus not only on the technical aspects of BGA interpretation but also on integrating this knowledge into clinical decision-making. Future research should evaluate the effectiveness of these interventions in improving interpretative accuracy, reducing errors, and ultimately enhancing patient outcomes. Additionally, future studies should evaluate BGA interpretation within the clinical context, considering the pressures of the clinical environment and assessing the accuracy and safety of BGA interpretation under these conditions.

## 5. Conclusions

This study has demonstrated that ED and ICU nurses are capable of effectively and safely ruling out patients who do not present with time-dependent conditions or require invasive therapy. However, the findings also reveal that nurses’ overall performance in identifying critical conditions remains suboptimal, underscoring the urgent need to address gaps in clinical competencies.

To bridge these gaps, healthcare institutions and policymakers should prioritize the development and implementation of structured, evidence-based training programs aimed at enhancing the interpretative and decision-making skills of critical care nurses. These programs should integrate simulation-based learning, multidisciplinary collaboration, and decision-support tools to better prepare nurses for high-stakes clinical environments.

Furthermore, future research should evaluate the long-term effectiveness of such educational interventions and explore their integration into standard nursing curricula. By addressing these areas, healthcare systems can promote safer and more efficient patient care, while empowering nurses to take on a more active role in complex clinical decision-making.

## Figures and Tables

**Table 1 healthcare-13-00261-t001:** Characteristics of the nurses enrolled in the study, divided between ICU and ED.

**Variables**	**ICU Nurse**	**ED Nurse**	***p*-Value**
Number of nurses	26	17	
Sex, n (%) Female Male	20 (76.9) 6 (23.1)	9 (52.9) 8 (47.1)	0.101
Age, mean (SD)	39.1 (11.1)	31.7 (7.6)	0.021
Total years of nursing experience, median (IQR)	10 (4–25)	5 (3–10)	0.022
Years worked in the current department, median (IQR)	8 (3–21)	4 (2–9)	0.051
Personal interest in the topic of BGA, mean (SD)	7.7 (2.1)	7.8 (1.3)	0.873
Self-reported engagement in personal study of BGA, n (%)	17 (65.4)	14 (87.5)	0.113
Self-assessed confidence in performing arterial puncture, median (IQR)	8 (2–10)	8 (8–9)	0.637
Self-assessed confidence in interpreting BGA results, mean (SD)	5.3 (1.5)	6.3 (0.9)	0.028
Self-reported frequency of performing BGA during a shift, median (IQR)	10 (10–10)	8 (7–9)	<0.001
Participation in a BGA-related training course within the past 5 years, n (%) No Yes	21 (80.8) 5 (19.2)	9 (52.9) 8 (47.0)	0.052
Highest degree obtained, n (%) Bachelor in Nursing Clinical Master’s Degree Master of Science in Nursing	24 (92.3) 0 (0.0) 2 (7.7)	15 (88.2) 1 (5.9) 1 (5.9)	0.450

**Table 2 healthcare-13-00261-t002:** Contingency tables comparing the urgency defined by physicians and that defined by ICU and ED nurses, with Cohen’s kappa reported at the end.

**ICU Nurse Urgency Level**	**Physicians Urgency Level**			
	Non-Urgent	Urgent	Emergency	Total, n (%)
Non-urgent	98	21	4	123 (29.5)
Urgent	104	58	25	187 (45.0)
Emergency	32	51	23	106 (25.5)
Total, n (%)	234 (56.3)	130 (31.2)	52 (12.5)	416 (100)
Cohen’s Kappa: 0.139 (95% CI: 0.083–0.178)				
**ED Nurse urgency level**	**Physician urgency level**			
	Non-urgent	Urgent	Emergency	Total, n (%)
Non-urgent	76	22	4	102 (37.5)
Urgent	68	37	6	111 (40.8)
Emergency	9	26	24	59 (21.7)
Total, n (%)	153 (56.2)	85 (31.3)	34 (12.5)	272 (100)
Cohen’s Kappa: 0.218 (95% CI: 0.203–0.260)				

**Table 3 healthcare-13-00261-t003:** The 2 × 2 contingency table evaluating the nurses’ ability in assessing the secondary outcomes of the study.

**Non-Invasive Therapy According to ICU Nurse**	**Non-Invasive Therapy According to Physicians**		**Non-Invasive Therapy According to ED Nurse**	**Non-Invasive Therapy According to Physicians**	
	No	Yes		No	Yes
No	44	12	No	68	18
Yes	190	170	Yes	85	101
Sensitivity: 93.4% (91.0–95.7) Specificity: 18.8 (15.0–22.6) PPV: 47.2% (42.4–52.0) NPV: 78.5% (74.6–82.5)			Sensitivity: 84.8% (80.6–89.1) Specificity: 44.4% (38.5–50.3) PPV: 54.3% (48.4–60.2) NPV: 79.1% (74.2–83.9)		
**Organ replacement therapy according to ICU Nurse**	**Organ replacement therapy according to Physicians**		**Organ replacement therapy according to ED Nurse**	**Organ replacement therapy according to Physicians**	
	No	Yes		No	Yes
No	260	64	No	194	62
Yes	50	38	Yes	9	6
Sensitivity: 37.2% (32.6–41.9) Specificity: 83.7% (80.3–87.4) PPV: 43.2% (38.4–47.9) NPV: 80.2% (76.4–84.1)			Sensitivity: 8.8% (5.4–12.2) Specificity: 95.6% (93.1–98.0) PPV: 40.0% (34.2–45.8) NPV: 75.8% (70.7–80.9)		

**Table 4 healthcare-13-00261-t004:** The 2 × 2 contingency table evaluating the ability of nurses from the two departments in relation to the primary outcome of the study.

	**Patients with Time-Dependent Medical Conditions**		
**Patients with time-dependent medical conditions according to ICU nurse**	No	Yes	Sensitivity: 43.3% (38.5–48.0) Specificity: 80.4% (76.6–84.2) PPV: 42.4% (37.7–47.2) NPV:80.9% (77.2–84.7)
No	251	59	
Yes	61	45	
Cohen’s kappa: 0.236 (95% CI: 0.133–0.339)			
**Patients with time-dependent medical conditions according to ED nurse**	**Patients with time-dependent medical conditions**		
	No	Yes	Sensitivity: 48.5% (42.6–54.5) Specificity: 87.2% (83.3–91.2) PPV: 55.9% (50.0–61.8) NPV: 83.6% (79.1–87.9)
No	178	35	
Yes	26	33	
Cohen’s kappa: 0.374 (95% CI: 0.247–0.502)			

**Table 5 healthcare-13-00261-t005:** Comparison of nurse characteristics, specifically analyzing those who made more than four errors in identifying patients with time-dependent medical conditions or incorrectly classified patients without time-dependent medical conditions as such. Groups are categorized based on the mean number of errors per nurse.

	**Time-Dependent Medical Conditions**		
**Variable**	**≤4 Errors**	**>4 Errors**	***p*-Value**
Number of nurses, n (%)	32 (74.4)	11 (25.6)	
Sex, n (%) Female Male	20 (62.5) 12 (37.8)	9 (81.8) 2 (18.2)	0.238
Age, mean (SD)	36.6 (10.4)	34.8 (10.6)	0.619
Total years of nursing experience, median (IQR)	10 (4–19)	4 (3–30)	0.329
Years worked in the current department, median (IQR)	5 (3–17)	4 (2–16)	0.586
Personal interest in the topic of BGA, mean (SD)	7.7 (1.6)	7.9 (2.4)	0.769
Self-reported engagement in personal study of BGA, n (%)	23 (74.2)	8 (72.7)	0.924
Self-assessed confidence in performing arterial puncture, median (IQR)	8 (7–10)	9 (2–10)	0.932
Self-assessed confidence in interpreting BGA results, mean (SD)	6.0 (1.2)	4.8 (1.6)	0.011
Self-reported frequency of performing BGA during a shift, median (IQR)	9 (8–10)	10 (8–10)	0.398
Participation in a BGA-related training course within the past 5 years, n (%) No Yes	18 (69.2) 8 (30.8)	14 (82.3) 3 (17.6)	0.335
Highest degree obtained, n (%) Bachelor in Nursing Clinical Master’s Degree Master of Science in Nursing	28 (87.5) 1 (3.1) 3 (9.4)	11 (100.0) 0 (0.0) 0 (0.0)	0.469
Working department, n (%) Emergency Department ICU	14 (43.7) 18 (56.2)	3 (27.3) 8 (72.7)	0.335

## Data Availability

Data available on request due to privacy/ethical restrictions.

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
