# Peer review of "Arterial Blood Gas Analysis and Clinical Decision-Making in Emergency and Intensive Care Unit Nurses: A Performance Evaluation"

_healthcare, 2025, doi:10.3390/healthcare13030261_

Round 1
Reviewer 1 Report
Comments and Suggestions for Authors
Dear Authors,
Please find attached my comment.

Reviewer 2 Report
Comments and Suggestions for Authors
Arterial Blood Gas Analysis and Clinical Decision-Making in Emergency and Intensive Care Unit Nurses: A Performance Evaluation
First of all,, I would like to thank you for the opportunity to review this research work. Below, I have outlined some suggestions for improving your article. I hope this feedback will help make your work ready for publication and aid in its dissemination, contributing to the scientific community.
Abstract: The abstract clearly describes the objectives, methods, results, and conclusions. However:
Consider removing redundancies, especially in the results section (e.g., "high specificity" and "sensitivity below 50%").
Emphasize the practical or policy implications of the findings more strongly.
Ensure the abstract complies with the word count limit.
Introduction: Clarify the gaps in the literature that the study aims to address. This is crucial for establishing the relevance of the research.
Avoid excessive use of consecutive references without integrating them into the narrative (e.g., "[2, 3, 4, 5]" in lines 42–47).
Methods: Explain in more detail how the clinical vignettes were validated to ensure their relevance.
Provide additional details on how the sample size was calculated. This will help readers better understand the study's scope and relevance.
Clarify whether quality control measures or standardization protocols were applied during questionnaire supervision. If so, describe how this was implemented.
Results: Highlight key findings to avoid overwhelming the reader with excessive details.
Provide more qualitative or narrative analysis to complement the data presented in the tables. For instance, at the beginning of the results section, expand on the context and summarize the key takeaways from the tables. This would strengthen the transition to the Discussion section.
Review the tables for formatting improvements, such as adjusting font sizes and centering items for better readability.
Discussion: Delve deeper into the study limitations (e.g., the simulated environment and its applicability to real clinical practice).
Propose specific interventions to address the identified skill gaps among nurses.
Conclusion: Include a more direct and concrete call to action regarding policies, education, or future research stemming from this study.
Avoid overly generic statements, such as "further research is needed."
Ethical Considerations: Please provide more details on how informed consent was ensured for the secondary data used in the study.
Thank you very much in advance for considering these suggestions. I hope they are helpful in improving your article and ensuring its publication success.
Reviewer 3 Report
Comments and Suggestions for Authors
The article effectively highlights the importance of ABG analysis in guiding clinical decisions for critically ill patients and addresses the gap in literature regarding the interpretive skills of emergency and ICU nurses. The title is clear, informative, and appropriately reflects the study's scope. The abstract succinctly summarizes the research background, methodology, primary and secondary outcomes, and key conclusions. Also, the structuring of primary and secondary outcomes aligns well with the study's objectives, offering a comprehensive framework for evaluating nurse competencies in ABG interpretation and decision-making.
While the article is a valuable contribution to the field, several critical areas require further refinement:
Firstly, the manuscript lacks clarity regarding the methods or frameworks nurses were trained in for ABG evaluation. Specify whether the Handerson-Hesselbach approach, Stewart's model, or a biophysicochemical perspective was applied.
Secondly, the participant size appears relatively small given the study's scope and target population. A power analysis is necessary to justify the sample size and demonstrate its adequacy for the statistical analyses performed.
Thirdly, Table 1 reveals significant differences in work experience between ICU and ED nurses. This factor could substantially influence study results. Despite these disparities, the outcomes between ICU and ED nurses are remarkably similar. Address this observation in the discussion and emphasize the critical role of continuous education and training, even for experienced personnel, to mitigate interpretative deficiencies.
Also, the presence of Italian text (yes/si) in Table 3 instead of the appropriate data is a notable oversight. Correct this error to ensure the table accurately reflects the study's findings.
Another point, expand the discussion to elaborate on the implications of the study's findings for clinical practice. Specifically, detail how targeted training programs can address the deficiencies observed in nurse interpretive skills and decision-making capabilities. Additionally, discuss how the findings could inform future protocols and curricula for nursing education.
Finally, address minor typographical issues throughout the manuscript. For instance, "Dara curation" should be corrected to "Data curation."
Reviewer 4 Report
Comments and Suggestions for Authors
Dear authors,
Than you.
I appreciate the effort put into this research and I believe that with the suggested refinements the article could have an even greater impact on the field:
Introduction
The introduction of the article clearly highlights the importance of blood gas analysis (BGA) in clinical practice, particularly in the care of critically ill patients. However, expanding the introduction by an additional half-page could provide a clearer picture, especially in terms of clarifying whether BGA interpretation is within the competencies of nurses and summarizing what other authors have written and concluded on this topic. Additionally, a more detailed theoretical review of the literature related to the study's subject is missing.
Methods and Results
The methods and results in this article are well-structured, detailed, and clearly defined contributing to the credibility and reproducibility of the research.
Discussion
Each finding (significant or non-significant) should be commented on and compared with previous studies. This would expand the discussion, which is currently very modest. The research does not delve into the specific interpretative skills that could be further explored for a better understanding of the problem. I also mentioned this in the introduction: the focus on specific competencies is limited. For example, in my country, nurses are not required to be familiar with, analyze or interpret these results. Their role is to draw blood, deliver it to the lab and notify the physician when the results are ready. They do not make decisions based on the results.
Limitations
The limitations should emphasize that the research was conducted in only two hospitals in Italy, which may restrict the generalizability of the results to broader populations or different healthcare systems. Although the vignettes reflect real-life cases, they cannot fully replicate the stress and complexity of actual clinical environments. These are all limitations that should be commented on.
Thank you once again!
Round 2
Reviewer 1 Report
Comments and Suggestions for Authors
Thanks for getting back to me and thinking about my ideas.
No more questions.
But there are still practical things the study doesn't look at enough, like safety, how efficient it is, and how quickly nurses can interpret ABGs, especially in tricky situations where accuracy is a must..
Reviewer 2 Report
Comments and Suggestions for Authors
Dear Authors,
I would like to express my gratitude for the responses provided to all the questions raised and for expanding on the requested information. This demonstrates a remarkable commitment to clarity and scientific rigor.
The only inconvenience I observe, and I understand this is a decision of the author, is the number of consecutive citations in the first paragraphs of the introduction, which might slightly hinder the initial reading flow.
Beyond that, I would like to congratulate you on your excellent work and the effort reflected in this manuscript. I wish you great success in the dissemination of your findings.
Sincerely
